# Improved Neural Relation Detection for Knowledge Base Question Answering

## Abstract

Relation detection is a core component for many NLP applications including Knowledge Base Question Answering (KBQA). In this paper, we propose a hierarchical recurrent neural network enhanced by residual learning that detects KB relations given an input question. Our method uses deep residual bidirectional LSTMs to compare questions and relation names via different hierarchies of abstraction. Additionally, we propose a simple KBQA system that integrates entity linking and our proposed relation detector to enable one enhance another. Experimental results evidence that our approach achieves not only outstanding relation detection performance, but more importantly, it helps our KBQA system to achieve state-of-the-art accuracy for both single-relation (SimpleQuestions) and multi-relation (WebQSP) QA benchmarks.

## 1 Introduction

Knowledge Base Question-Answering (KBQA) systems answer questions by obtaining information from KB tuples (Berant et al., 2013; Yao et al., 2014; Bordes et al., 2015; Bast and Haussmann, 2015; Yih et al., 2015; Xu et al., 2016). For an input question, these systems typically generate a KB query, which can be executed to retrieve the answers from a KB. Figure 1 illustrates the process used to parse two sample questions in a KBQA system: (a) a single-relation question, which can be answered with a single <*head-entity, relation, tail-entity*> KB tuple (Fader et al., 2013; Yih et al., 2014; Bordes et al., 2015); and (b) a more complex case, where some constraints need to be handled for multiple entities in the question. The KBQA

system in the figure performs two key tasks: (1) *entity linking*, which links $n$-grams in questions to KB entities, and (2) *relation detection*, which identifies the KB relation(s) a question refers to.

The main focus of this work is to improve the *relation detection* subtask and further explore how it can contribute to the KBQA system. Although general relation detection[1] methods are well studied in the NLP community, such studies usually do not take the end task of KBQA into consideration. As a result, there is a significant gap between general relation detection studies and KB-specific relation detection. First, in most general relation detection tasks, the number of target relations is limited, normally smaller than 100. In contrast, in KBQA even a small KB, like Freebase2M (Bordes et al., 2015), contains more than 6,000 relation types. Second, relation detection for KBQA often becomes a zero-shot learning task, since some golded test relations may not appear in the training data. For example, the SimpleQuestions (Bordes et al., 2015) data set has 14% of the golden test relations not observed in golden training tuples. Third, as shown in Figure 1(b), for some KBQA tasks like WebQuestions (Berant et al., 2013), we need to predict a chain of relations instead of a single relation. This increases the number of target relation types and the sizes of candidate relation pools, further increasing the difficulty of KB relation detection. Owing to these reasons, KB relation detection is significantly more challenging compared to general relation detection tasks.

This paper improves KB relation detection to cope with the above mentioned problems. First, in order to deal with the unseen relations, we propose to break the relation names into word sequences for question-relation matching. Second, noticing that original relation names can sometimes help

---

[1] In the information extraction field such tasks are usually called *relation extraction* or *relation classification*.

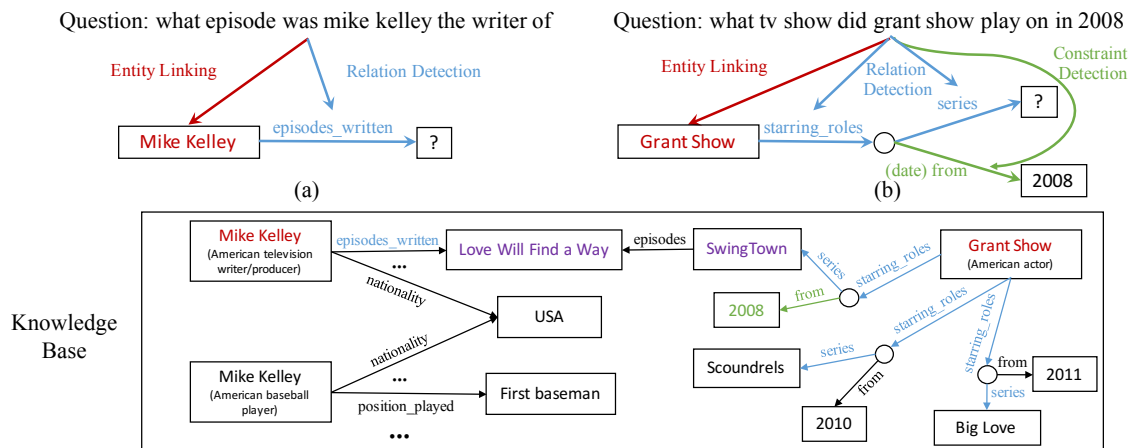

Figure 1: KBQA examples and its three key components. (a) A single relation example. We first identify the topic entity with *entity linking* and then detect the relation asked by the question with *relation detection* (from all relations connecting the topic entity). Based on the detected entity and relation, we form a query to search the KB for the correct answer "*Love Will Find a Way*". (b) A more complex question containing two entities. By using "*Grant Show*" as the topic entity, we could detect a chain of relations "*starring_roles-series*" pointing to the answer. An additional *constraint detection* takes the other entity "*2008*" as a constraint, to filter the correct answer "*SwingTown*" from all candidates found by the topic entity and relation.

to match longer question contexts, we propose to build both relation-level and word-level relation representations. Third, we use deep bidirectional LSTMs (*BiLSTM*s) to learn different levels of question representations in order to match the different levels of relation information. Finally, we propose a residual learning method for sequence matching, which makes the model training easier and results in more abstract (deeper) question representations, thus improves hierarchical matching.

In order to assess how the proposed *improved relation detection* could benefit the KBQA end task, we also propose a simple KBQA implementation composed of *two-step relation detection*. Given an input question and a set of candidate entities retrieved by an entity linker based on the question, our proposed relation detection model plays a key role in the KBQA process: (1) Re-ranking the entity candidates according to whether they connect to high confident relations detected from the *raw question text* by the relation detection model. This step is important to deal with the ambiguities normally present in entity linking results. (2) Finding the core relation (chains) for each *topic entity*[2] selection from a much smaller candidate entity set after re-ranking. The above steps are followed by an optional constraint detection step, when the question cannot be answered by single relations (e.g. containing multiple entities). Finally the highest scored query from the above steps is used to query the KB for answers.

Our main contributions include: (i) An improved relation detection model by hierarchical matching between questions and relations with residual learning; (ii) We demonstrate that the improved relation detector enables our simple KBQA system to achieve state-of-the-art results on both single-relation and multi-relation KBQA tasks.

## 2 Related Work

**Relation Extraction** Relation extraction (RE) is an important sub-field of information extraction. General research in this field usually works on a (small) pre-defined relation set, where given a text paragraph and two target entities, the goal is to determine whether the text indicates any types of relations between the entities or not. As a result RE is usually formulated as a **classification task**. Traditional RE methods rely on large amount of hand-crafted features (Zhou et al., 2005; Rink and Harabagiu, 2010; Sun et al., 2011). Recent research benefits a lot from the advancement of deep learning: from word embeddings (Nguyen and Grishman, 2014; Gormley et al., 2015) to deep networks like CNNs and LSTMs (Zeng et al., 2014; dos Santos et al., 2015; Vu et al., 2016) and attention models (Zhou et al., 2016; Wang et al., 2016).

The above research assumes there is a fixed (closed) set of relation types, thus no zero-shot learning capability is required. The number of relations is usually not large: The widely used ACE2005 has 11/32 coarse/fine-grained relations; SemEval2010 Task8 has 19 relations; TAC-KBP2015 has 74 relations although it considers

---

[2]Following (Yih et al., 2015), here *topic entity* refers to the node of the (directed) query tree; and *core-chain* is the directed path of relation from root to the answer node.

open-domain Wikipedia relations. All are much fewer than thousands of relations in KBQA. As a result, few work in this field focuses on dealing with large number of relations or unseen relations.

**Relation Detection in KBQA Systems** Relation detection for KBQA also starts with feature-rich approaches (Yao and Van Durme, 2014; Bast and Haussmann, 2015) towards usages of deep networks (Yih et al., 2015; Xu et al., 2016; Dai et al., 2016) and attention models (Yin et al., 2016; Golub and He, 2016). Many of the above relation detection research could naturally support large relation vocabulary and open relation sets (especially for QA with OpenIE KB like ParaLex (Fader et al., 2013)), in order to fit the goal of open-domain question answering.

Different KBQA data sets have different levels of requirement about the above open-domain capacity. For example, most of the gold test relations in WebQuestions can be observed during training, thus some prior work on this task adopted the close domain assumption like in the general RE research. While for data sets like SimpleQuestions and ParaLex, the capacity to support large relation sets and unseen relations becomes more necessary. To the end, there are two main solutions: (1) use pre-trained relation embeddings (e.g. from TransE (Bordes et al., 2013)), like (Dai et al., 2016); (2) factorize the relation names to sequences and formulate relation detection as a **sequence matching and ranking** task. Such factorization works because that the relation names usually comprise meaningful word sequences. For example, Yin et al. (2016) split relations to word sequences for single-relation detection. Liang et al. (2016) also achieve good performance on WebQSP with word-level relation representation in an end-to-end neural programmer model. Yih et al. (2015) use character tri-grams as inputs on both question and relation sides. Golub and He (2016) propose a generative framework for single-relation KBQA which predicts relation with a character-level sequence-to-sequence model.

Another difference between relation detection in KBQA and general RE is that general RE research assumes that the two argument entities are both available. Thus it usually benefits from features (Nguyen and Grishman, 2014; Gormley et al., 2015) or attention mechanisms (Wang et al., 2016) based on the entity information (e.g. entity types or entity embeddings). For relation detec-

tion in KBQA, such information is mostly missing because: (1) one question usually contains single argument (the topic entity) and (2) one KB entity could have multiple types (type vocabulary size larger than 1,500). This makes KB entity typing itself a difficult problem so no previous used entity information in the relation detection model.[3]

## 3 Improved KB Relation Detection

### 3.1 Different Granularity in KB Relations

Previous research formulates KB relation detection as a sequence matching problem. However, while the questions are natural word sequences, how to represent relations as sequences remains a challenging problem. Here we give an overview of two types of relation sequence representations commonly used in previous work:

**(1) Relation Name as a Single Token** (*relation-level*). In this case, each relation name is treated as a unique token. The problem with this approach is that it suffers from the low relation coverage due to limited amount of training data, thus cannot generalize well to large number of open-domain relations. For example in Figure 1, when treating relation names as single tokens, it will be difficult to match the questions to relation names "*episodes_written*" and "*starring_roles*" if these names do not appear in training data – their relation embeddings $\mathbf{h}^r$s will be random vectors thus are not comparable to question embeddings $\mathbf{h}^q$s.

**(2) Relation as Word Sequence** (*word-level*). In this case, the relation is treated as a sequence of words from the tokenized relation name. It has better generalization, but suffers from the lack of global information from the original relation names. For example in Figure 1(b), when doing only word-level matching, it is difficult to rank the target relation "*starring_roles*" higher compared to the incorrect relation "*plays_produced*". This is because the incorrect relation contains word "*plays*", which is more similar to the question (containing word "*play*") in the embedding space. On the other hand, if the target relation co-occurs with questions related to "*tv appearance*" in training, by treating the whole relation as a token (i.e. relation id), we could better learn the correspondence between this token and phrases like "*tv show*" and "*play on*".

---

[3] Such entity information has been used in KBQA systems as features for the final answer re-rankers.

| | Relation Token | Question 1 | Question 2 |
|---|---|---|---|
| | | what tv episodes were <e> the writer of | what episode was written by <e> |
| **relation-level** | episodes_written | *tv episodes were <e> the writer of* | *episode was written by <e>* |
| **word-level** | episodes | *tv episodes* | *episode* |
| | written | *the writer of* | *written* |

Table 1: An example of KB relation (*episodes_written*) with two types of relation tokens (relation names and words), and two questions asking this relation. The topic entity is replaced with token <e> which could give the position information to the deep networks. The italics show the evidence phrase for each relation token in the question.

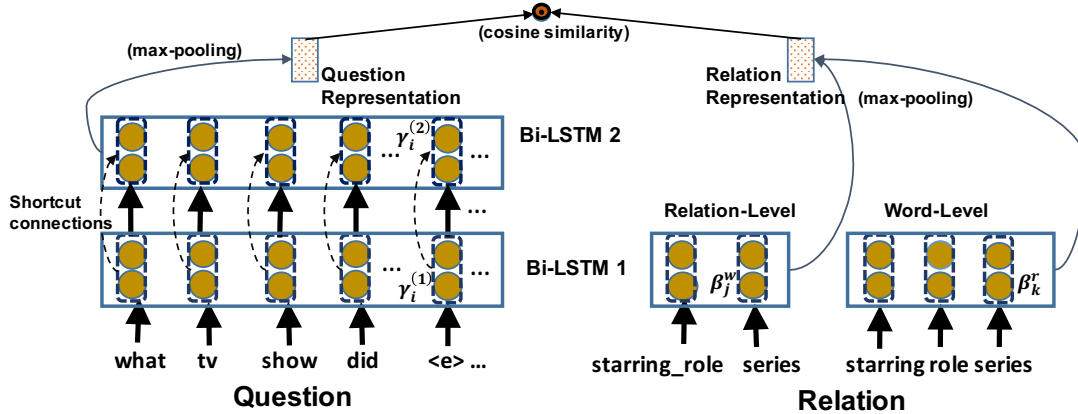

Figure 2: The proposed Hierarchical Residual BiLSTM (HR-BiLSTM) model for relation detection. Note that without the dotted arrows of shortcut connections between two layers, the model will only compute the similarity between the second-layer of questions representations and the relation, thus is not doing hierarchical matching.

The two types of relation representation contain different levels of abstraction. As shown in Table 1, the word-level focus more on local information (words and short phrases), and the relation-level focus more on global information (long phrases and skip-grams) but suffer from data sparsity. Since the two levels of granularity both have their own pros and cons, we propose a hierarchical matching approach for KB relation detection: for a candidate relation, our approach matches the input question to both word-level and relation-level representations to get the final ranking score. Section 3.2 gives the details of our proposed approach.

### 3.2 Hierarchical Sequence Matching with Residual Learning for Relation Detection

This section describes our hierarchical matching approach. In order to match the question to different aspects of a relation (with different abstraction levels), we need to deal with the following problems on learning question/relation representations.

**Relation Representations from Different Granularity.** We provide our model with both types of relation representation: word-level and relation-level. Therefore, the input relation becomes $\mathbf{r} = \{r_1^{word}, \cdots, r_{M_1}^{word}\} \cup \{r_1^{rel}, \cdots, r_{M_2}^{rel}\}$, where the first $M_1$ tokens are words (e.g. {*episode, written*}), and the last $M_2$ tokens are relation names, e.g., {*episode_written*} or {*starring_roles, series*}

(when the target is a chain like in Figure 1(b)). We transform each token above to its word embedding then use two BiLSTMs (with shared parameters) to get their hidden representations $[\mathbf{B}_{1:M_1}^{word} : \mathbf{B}_{1:M_2}^{rel}]$ (each row vector $\boldsymbol{\beta}_i$ is the concatenation between forward/backward representations at $i$). We initialize the relation sequence LSTMs with the final state representations of the word sequence, as a back-off for unseen relations. We apply *one* max-pooling on these two sets of vectors and get the final relation representation $\mathbf{h}^r$.

**Different Abstractions of Questions Representations.** From Table 1, we can see that different parts of a relation could match different contexts of question texts. Usually relation names could match longer phrases in the question and relation words could match short phrases. Yet different words might match phrases of different lengths.

As a result, we hope the question representations could also comprise vectors that summarize various lengths of phrase information (different levels of abstraction), in order to match relation representations of different granularity. We deal with this problem by applying deep BiLSTMs on questions. The first-layer of BiLSTM works on the word embeddings of question words $\mathbf{q} = \{q_1, \cdots, q_N\}$ and gets hidden representations $\boldsymbol{\Gamma}_{1:N}^{(1)} = [\boldsymbol{\gamma}_1^{(1)}; \cdots; \boldsymbol{\gamma}_N^{(1)}]$. The second-layer BiLSTM works on $\boldsymbol{\Gamma}_{1:N}^{(1)}$ to get the second set of hid-

den representations $\Gamma_{1:N}^{(2)}$. Since the second BiL-STM starts with the hidden vectors from the first layer, intuitively it could learn more general and abstract information compared to the first layer.

**Hierarchical Matching between Relation and Question**   Now we have question contexts of different lengths encoded in $\Gamma_{1:N}^{(1)}$ and $\Gamma_{1:N}^{(2)}$. Unlike the standard usage of deep BiLSTMs that employs the representations in the final layer for prediction, here we expect that two layers of question representations can be complementary to each other and both should be compared to the relation representation space (*Hierarchical Matching*). This is important for our task since each relation token can correspond to phrases of different lengths, mainly because of syntactic variations. For example in Table 1, the relation word *written* could be matched to either the same single word in the question or a much longer phrase *be the writer of*.

We could perform the above hierarchical matching by computing the similarity between each layer of $\Gamma$ and $\mathbf{h}^r$ separately and doing the (weighted) sum between the two scores. However this does not give significant improvement (see Table 2). Our analysis shows that this naive method suffers from the training difficulty, evidenced by that the converged training loss of this model is much higher than that of a single-layer baseline model. This is mainly because (1) Deep BiLSTMs do not guarantee that the two-levels of question hidden representations are comparable, the training usually falls to a local optima where one layer has good matching scores and the other always has weight close to 0. (2) The training of deeper architectures itself is more difficult.

To overcome the above difficulties, we adopt the idea from Residual Networks (He et al., 2016) for hierarchical matching by adding shortcut connections between two BiLSTM layers. We proposed two ways of such *Hierarchical Residual Matching*: (1) Connecting each $\gamma_i^{(1)}$ and $\gamma_i^{(2)}$, resulting in a $\gamma_i' = \gamma_i^{(1)} + \gamma_i^{(2)}$ for each position $i$. Then the final question representation $\mathbf{h}^q$ becomes a max-pooling over all $\gamma_i'$s, $1 \leq i \leq N$. (2) Applying max-pooling on $\Gamma_{1:N}^{(1)}$ and $\Gamma_{1:N}^{(2)}$ to get $\mathbf{h}_{max}^{(1)}$ and $\mathbf{h}_{max}^{(2)}$, respectively, then setting $\mathbf{h}^q = \mathbf{h}_{max}^{(1)} + \mathbf{h}_{max}^{(2)}$. Finally we compute the matching score of $\mathbf{r}$ given $\mathbf{q}$ as $s_{rel}(\mathbf{r}; \mathbf{q}) = cos(\mathbf{h}^r, \mathbf{h}^q)$.

Intuitively, the proposed method should benefit from hierarchical training since the second layer is fitting the residues from the first layer of matching, so the two layers of representations are more likely to be complementary to each other. This also ensures the vector spaces of two layers are comparable and makes the second-layer training easier.

During training we adopt a ranking loss to maximizing the margin between the gold relation $\mathbf{r}^+$ and other relations $\mathbf{r}^-$ in the candidate pool $R$.

$$l_{\text{rel}} = \max\{0, \gamma - s_{\text{rel}}(\mathbf{r}^+; \mathbf{q}) + s_{\text{rel}}(\mathbf{r}^-; \mathbf{q})\}$$

where $\gamma$ is a constant parameter. Fig 2 summarizes the above *Hierarchical Residual BiLSTM (HR-BiLSTM)* model.

**Remark:** Another way of hierarchical matching consists in relying on **attention mechanism**, e.g. (Parikh et al., 2016), to find the correspondence between different levels of representations. This performs below the HR-BiLSTM (see Table 2).

# 4   KBQA Enhanced by Relation Detection

This section describes our KBQA pipeline system. We make minimal efforts beyond the training of the relation detection model, making the whole system easy to build.

Following previous work (Yih et al., 2015; Xu et al., 2016), our KBQA system takes an existing entity linker to produce the top-$K$ linked entities, $EL_K(q)$, for a question $q$ ("*initial entity linking*"). Then we generate the KB queries for $q$ following the four steps illustrated in Algorithm 1.

---

**Algorithm 1:** KBQA with two-step relation detection

**Input**   : Question $q$, Knowledge Base $KB$, the initial top-$K$ entity candidates $EL_K(q)$
**Output**: Top query tuple $(\hat{e}, \hat{r}, \{(c, r_c)\})$

1  **Entity Re-Ranking** (*first-step relation detection*): Use the *raw question text* as input for a relation detector to score all relations in the KB that are associated to the entities in $EL_K(q)$; use the relation scores to re-rank $EL_K(q)$ and generate a shorter list $EL'_{K'}(q)$ containing the top-$K'$ entity candidates (Section 4.1)

2  **Relation Detection**: Detect relation(s) using the *reformatted question text* in which the topic entity is replaced by a special token $<e>$ (Section 4.2)

3  **Query Generation**: Combine the scores from step 1 and 2, and select the top pair $(\hat{e}, \hat{r})$ (Section 4.3)

4  **Constraint Detection** (optional): Compute similarity between $q$ and any neighbor entity $c$ of the entities along $\hat{r}$ (connecting by a relation $r_c$), add the high scoring $c$ and $r_c$ to the query (Section 4.4).

---

Compared to previous approaches, the main difference is that we have an additional *entity re-ranking* step after the *initial entity linking*. We have this step because we have observed that entity

linking sometimes becomes a bottleneck in KBQA systems. For example, on SimpleQuestions the best reported linker could only get 72.7% top-1 accuracy on identifying topic entities. This is usually due to the ambiguities of entity names, e.g. in Fig 1(a), there are *TV writer* and *baseball player* "*Mike Kelley*", which is impossible to distinguish with only entity name matching.

Having observed that different entity candidates usually connect to different relations, here we propose to help entity disambiguation in the *initial entity linking* with relations detected in questions.

Sections 4.1 and 4.2 elaborate how our relation detection help to re-rank entities in the initial entity linking, and then those re-ranked entities enable more accurate relation detection. The KBQA end task, as a result, benefits from this process.

## 4.1 Entity Re-Ranking

In this step, we use the *raw question text* as input for a relation detector to score all relations in the KB with connections to at least one of the entity candidates in $EL_K(q)$. We call this step *relation detection on entity set* since it does not work on a single topic entity as the usual settings. We use the HR-BiLSTM as described in Sec. 3. For each question $q$, after generating a score $s_{rel}(r; q)$ for each relation using HR-BiLSTM, we use the top $l$ best scoring relations $(R_q^l)$ to re-rank the original entity candidates. Concretely, for each entity $e$ and its associated relations $R_e$, given the original entity linker score $s_{linker}$, and the score of the most confident relation $r \in R_q^l \cap R_e$, we sum these two scores to re-rank the entities:

$$s_{\text{rerank}}(e; q) = \alpha \cdot s_{\text{linker}}(e; q)$$
$$+ (1 - \alpha) \cdot \max_{r \in R_q^l \cap R_e} s_{\text{rel}}(r; q).$$

Finally, we select top $K' < K$ entities according to score $s_{rerank}$ to form the re-ranked list $EL'_{K'}(q)$.

We use the same example in Fig 1(a) to illustrate the idea. Given the input question in the example, a relation detector is very likely to assign high scores to relations such as "*episodes_written*", "*author_of*" and "*profession*". Then, according to the connections of entity candidates in KB, we find that the TV writer "*Mike Kelley*" will be scored higher than the baseball player "*Mike Kelley*", because the former has the relations "*episodes_written*" and "*profession*". This method can be viewed as exploiting entity-relation collocation for entity linking.

## 4.2 Relation Detection

In this step, for each candidate entity $e \in EL'_K(q)$, we use the question text as the input to a relation detector to score all the relations $r \in R_e$ that are associated to the entity $e$ in the KB.[4] Because we have a single topic entity input in this step, we do the following question reformatting: we replace the the candidate $e$'s entity mention in $q$ with a token "$<e>$". This helps the model better distinguish the relative position of each word compared to the entity. We use the HR-BiLSTM model to predict the score of each relation $r \in R_e$: $s_{rel}(r; e, q)$.

## 4.3 Query Generation

Finally, the system outputs the $<$entity, relation (or core-chain)$>$ pair $(\hat{e}, \hat{r})$ according to:

$$s(\hat{e}, \hat{r}; q) = \max_{e \in EL'_{K'}(q), r \in R_e} (\beta \cdot s_{\text{rerank}}(e; q)$$
$$+ (1 - \beta) \cdot s_{\text{rel}}(r; e, q)),$$

where $\beta$ is a hyperparameter to be tuned.

## 4.4 Constraint Detection

Similar to (Yih et al., 2015), we adopt an additional constraint detection step based on text matching. Our method can be viewed as entity-linking on a KB sub-graph. It contains two steps: (1) **Sub-graph generation**: given the top scored query generated by the previous 3 steps[5], for each node $v$ (answer node or the CVT node like in Figure 1(b)), we collect all the nodes $c$ connecting to $v$ (with relation $r_c$) with any relation, and generate a sub-graph associated to the original query. (2) **Entity-linking on sub-graph nodes**: we compute a matching score between each $n$-gram in the input question (without overlapping the topic entity) and entity name of $c$ (except for the node in the original query) by taking into account the maximum overlapping sequence of characters between them (see Appendix A for details and B for special rules dealing with date/answer type constraints). If the matching score is larger than a threshold $\theta$ (tuned on training set), we will add the constraint entity $c$ (and $r_c$) to the query by attaching it to the corresponding node $v$ on the core-chain.

---

[4]Note that the number of entities and the number of relation candidates will be much smaller than those in the previous step.

[5]Starting with the top-1 query suffers more from error propagation. However we still achieve state-of-the-art on WebQSP in Sec.5, showing the advantage of our relation detection model. We leave in future work beam-search and feature extraction on beam for final answer re-ranking like in previous research.

| Model | Relation Input Views | Accuracy | |
|---|---|---|---|
| | | SimpleQuestions | WebQSP |
| AMPCNN (Yin et al., 2016) | words | 91.3 | - |
| BiCNN (Yih et al., 2015) | char-3-gram | 90.0 | 77.74 |
| BiLSTM w/ words | words | 91.2 | 79.32 |
| BiLSTM w/ relation names | rel_names | 88.9 | 78.96 |
| Hier-Res-BiLSTM (HR-BiLSTM) | words + rel_names | **93.3** | **82.53** |
| w/o rel_name | words | 91.3 | 81.69 |
| w/o rel_words | rel_names | 88.8 | 79.68 |
| w/o residual learning (weighted sum on two layers) | words + rel_names | 92.5 | 80.65 |
| replacing residual with attention (Parikh et al., 2016) | words + rel_names | 92.6 | 81.38 |
| single-layer BiLSTM question encoder | words + rel_names | 92.8 | 78.41 |
| replacing BiLSTM with CNN (HR-CNN) | words + rel_names | 92.9 | 79.08 |

Table 2: Accuracy on the SimpleQuestions and WebQSP relation detection tasks (test sets). The top shows performance of baselines. On the bottom we give the results of our proposed model together with the ablation tests.

## 5 Experiments

### 5.1 Task Introduction & Settings

We use the SimpleQuestions (Bordes et al., 2015) and WebQSP (Yih et al., 2016) datasets. Each question in these datasets is labeled with the gold semantic parse. Hence we can directly evaluate the relation detection performance independently as well as evaluate on the KBQA end task.

**SimpleQuestions (SQ):** It is a single-relation KBQA task. The KB we use consists of a Freebase subset with 2M entities (FB2M) (Bordes et al., 2015), in order to compare with previous research. Yin et al. (2016) also evaluated their relation extractor on this data set and released their proposed question-relation pairs, so we run our relation detection model on their data set. For the KBQA evaluation, we also start with their entity linking results[6]. Therefore, our results can be compared with their reported results on both tasks.

**WebQSP (WQ):** A multi-relation KBQA task. For this evaluation we use the full Freebase KB. We downloaded the S-MART (Yang and Chang, 2015) entity-linking results from (Yih et al., 2016)[7]. In order to evaluate the relation detection models, we create a new relation detection task from the WebQSP data set [8]. For each question and its labeled semantic parse: (1) we first select the topic entity from the parse; and then (2) select all the relations and relation chains (length $\leq$ 2) connecting to the topic entity, and set the core-chain labeled in the parse as the positive label and all the others as the negative examples.

We tune the following hyper-parameters on development sets: (1) the size of hidden state for LSTMs ($\{50, 100, 200, 400\}$)[9]; (2) learning rate

($\{0.1, 0.5, 1.0, 2.0\}$); (3) whether the shortcut connections are between hidden states or between max-pooling results (see Section 3.2); (4) The number of training epochs.

For both the relation detection experiments and the second-step relation detection in KBQA, we have *entity replacement* first (see Section 4.2 and Figure 1). All words are initialized by 300-$d$ pre-trained word embeddings (Mikolov et al., 2013).

### 5.2 Relation Detection Results

Table 2 shows the results on two relation detection tasks. The AMPCNN result is from (Yin et al., 2016), which got the previous state-of-the-art by outperforming several attention-based methods. We re-implemented the BiCNN from (Yih et al., 2015), where both questions and relations are represented with the word hash trick on character tri-grams. The baseline BiLSTM with relation word sequence appears to be the best baseline on WebQSP and is close to the previous best result of AMPCNN on SimpleQuestions. Our proposed HR-BiLSTM outperformed the best baselines on both tasks by margins of 2-3% (p $<$ 0.001 and 0.01 compared to the best baseline *BiLSTM w/ words* on SQ and WQ respectively).

Note that in the BiLSTM baselines, using only relation-names causes a great performance drop on SimpleQuestions compared to using only relation-words (91.2 to 88.9), while the gap is small on WebQSP. This shows that the SimpleQuestions suffers more from unseen relations while WebQSP questions can usually be answered by a fixed small set of relations and relation chains.

**Ablation Test:** The bottom of Table 2 shows ablation results of the proposed HR-BiLSTM. First, both tasks benefit from hierarchical matching between questions and both relation names and relations words, especially for SimpleQuestions (93.3% vs 91.2/88.8%). Second, residual

---

[6]The two resources have been downloaded from https://github.com/Gorov/SimpleQuestions-EntityLinking

[7]https://github.com/scottyih/STAGG

[8]We will make the datasets publicly available soon.

[9]For CNNs we double the size for fair comparison.

learning helps hierarchical matching compared to weighted-sum and attention baselines (see Section 3.2). For the attention baseline we tried the model from (Parikh et al., 2016) and its one-way variations, where the one-way model gives better results[10]. Note that residual learning significantly helps on WebQSP (80.65% to 82.53%), while it does not help as much on SimpleQuestions. On SimpleQuestions, even removing the deep layers does not change much the performance. WebQSP benefits more from residual and deeper architecture, possibly because in this dataset it is more important to handle larger scope of context matching.

Finally, on WebQSP, replacing BiLSTM with CNN in our hierarchical matching framework results in a large performance drop. Yet on SimpleQuestions the gap is small. We believe this is because the LSTM relation-encoder can better learn the composition of chains of relations in WebQSP, as it is better at dealing with longer dependencies.

**Remark:** To verify that residual learning helps WebQSP because it helps the hierarchical architecture learn different levels of abstraction instead of only benefiting from combination of two BiLSTMs, we replace the deep BiLSTM question encoder with two single-layer BiLSTMs (both on words), with shortcut connections between their hidden states. This decreases accuracy to 76.11%, probably because of over-fitting (similar training accuracy compared to HR-BiLSTM). This proves that the residual and deep structures both contribute to the good performance of HR-BiLSTM.

### 5.3 KBQA End-Task Results

Table 3 compares our system with two published baselines (1) STAGG (Yih et al., 2015), the state-of-the-art on WebQSP[11] and (2) AMPCNN (Yin et al., 2016), the state-of-the-art on SimpleQuestions. Since these two baselines are specially designed/tuned for one particular dataset, they lead to limited results when applied to the other dataset. In order to highlight the effect of different relation detection models on the KBQA end-task, we also implemented another baseline that uses our KBQA system but replaces HR-BiLSTM with our implementation of AMPCNN (for SimpleQuestions) or the char-3-gram BiCNN (for WebQSP) relation detectors (second block in Table 3).

Compared to the *baseline relation detector* (3rd

---

[10]We think the idea of hierarchical matching with attention mechanism may work better for long sequences, and leave this for future work.

[11]The STAGG score on SQ is from (Bao et al., 2016).

| System | Accuracy | |
| | SQ | WQ |
| --- | --- | --- |
| STAGG | 72.8 | **63.9** |
| AMPCNN (Yin et al., 2016) | *76.4* | - |
| Baseline: Our Method w/ baseline relation detector | 75.1 | 60.0 |
| Our Method | **77.0** | 63.0 |
| w/o entity re-ranking | 74.9 | 60.6 |
| w/o constraints | - | 58.0 |
| Our Method (multi-detectors) | **78.7** | **63.9** |

Table 3: KBQA results on SimpleQuestions (SQ) and WebQSP (WQ) (test set). The numbers in *green* color are directly comparable to our results since we start with the same entity linking results.

row of results), our method, which includes an improved relation detector (HR-BiLSTM), improves from 2 to 3% the KBQA end task (4th row). Note that in contrast to previous KBQA systems, our system does not use joint-inference or feature-based re-ranking step, nevertheless it still achieves better or comparable results to the state-of-the-art.

The third block of the table details two ablation tests for the proposed components in our KBQA systems: (1) Removing the entity re-ranking step significantly decreases the scores. Since the re-ranking step relies on the relation detection models, this shows that our HR-BiLSTM model contributes to the good performance in multiple ways. Appendix C gives the detailed performance of the re-ranking step. (2) In contrast to the conclusion in (Yih et al., 2015), constraint detection is crucial for our system. This is probably because our joint performance on topic entity and core-chain detection is more accurate (77.5% top-1 accuracy), leaving a huge potential (77.5% vs. 58.0%) for the constraint detection module to improve.

Finally, like STAGG that uses multiple relation detectors (see (Yih et al., 2015) for the three models used), we also try to use the top-3 relation detectors from Section 5.2. As shown on the last row of Table 3, this gives a significant performance boost, resulting in a new state-of-the-art result on SimpleQuestions and a result comparable to the state-of-the-art on WebQSP.

## 6 Conclusion

KB relation detection is a key step in KBQA and has significant differences from general relation extraction tasks. We propose a novel KB relation detection model, HR-BiLSTM, that performs hierarchical matching between questions and KB relations. Our model outperforms the previous methods on KB relation detection tasks and allows our KBQA system to achieve state-of-the-arts.

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
