# Peer review of "Improved Neural Relation Detection for Knowledge Base Question Answering"

_ACL 2017 — decision unknown_

[Official Review · Reviewer 1 · rating 4 · confidence 4]
soundness 4 · originality 3 · clarity 4 · impact 3 · substance 5 · appropriateness 5 · meaningful comparison 2 · presentation format Oral Presentation

- Strengths: The paper addresses a relevant topic: learning the mapping between
natural language and KB relations, in the context of QA (where we have only
partial information for one of the arguments), and in the case of having a very
large number of possible target relations.

The proposal consists in a new method to combine two different representations
of the input text: a word level representation (i.e. with segmentation of the
target relation names and also the input text), and relations as a single token
(i.e without segmentation of relation names nor input text). 

It seems, that the main contribution in QA is the ability to re-rank entities
after the Entity Linking step.

Results show an improvement compared with the state of the art. 

- Weaknesses:
The approach has been evaluated in a limited dataset. 

- General Discussion:

I think, section 3.1 fits better inside related work, so the 3.2 can become
section 3 with the proposal. Thus, new section 3 can be splitted more properly.